# Ecological drift during colonization drives within-host and between-host heterogeneity in an animal-associated symbiont

**Jason Z. Chen**[1]*, **Zeeyong Kwong**[2], **Nicole M. Gerardo**[1], **Nic M. Vega**[1,3]

**1** Department of Biology, Emory University, Atlanta, Georgia, United States of America, **2** Laboratory of Bacteriology, National Institutes of Allergy and Infectious Diseases, Hamilton, Montana, United States of America, **3** Department of Physics, Emory University, Atlanta, Georgia, United States of America

* jche226@emory.edu

**Data Availability Statement:** Raw image files are deposited in the Emory Dataverse open repository (https://dataverse.unc.edu/dataset.xhtml?

## Abstract

Specialized host–microbe symbioses canonically show greater diversity than expected from simple models, both at the population level and within individual hosts. To understand how this heterogeneity arises, we utilize the squash bug, *Anasa tristis*, and its bacterial symbionts in the genus *Caballeronia*. We modulate symbiont bottleneck size and inoculum composition during colonization to demonstrate the significance of ecological drift, the noisy fluctuations in community composition due to demographic stochasticity. Consistent with predictions from the neutral theory of biodiversity, we found that ecological drift alone can account for heterogeneity in symbiont community composition between hosts, even when 2 strains are nearly genetically identical. When acting on competing strains, ecological drift can maintain symbiont genetic diversity among different hosts by stochastically determining the dominant strain within each host. Finally, ecological drift mediates heterogeneity in isogenic symbiont populations even within a single host, along a consistent gradient running the anterior-posterior axis of the symbiotic organ. Our results demonstrate that symbiont population structure across scales does not necessarily require host-mediated selection, as it can emerge as a result of ecological drift acting on both isogenic and unrelated competitors. Our findings illuminate the processes that might affect symbiont transmission, coinfection, and population structure in nature, which can drive the evolution of host–microbe symbioses and microbe–microbe interactions within host-associated microbiomes.

## Introduction

A persistent paradox in the study of host–microbe symbioses is that, like microbes in natural environments, microbial symbionts exhibit enormous strain diversity [1–8]. This is observed even when natural selection, imposed by specialized interactions with their hosts, is expected to erode genetic variation. Different mechanisms, based on environmental selection or host variation, are typically invoked to explain the maintenance of symbiont genetic variation, often in terms of host benefit [9,10]. However, these hypotheses do not account for how host-

persistentId=doi:10.15139/S3/YZPBGY). All other data and code are within the paper and its Supporting Information files.

**Funding:** This work was funded by Emory University and USDA NIFA 2019-67013-29371 to NMG. The funders had no role in study design, data collection and analysis, decision to publish, or preparation of the manuscript.

**Competing interests:** The authors have declared that no competing interests exist.

associated consortia assemble as ecological communities, which embeds this genetic variation within patches in physical space [11,12]. This is an inherently stochastic process that generates heterogeneity [13–15]. Heterogeneity in host-associated microbial communities manifests at 2 scales: as heterogeneity in colonization between hosts, and as spatial heterogeneity across tissues and organs within each host. At both scales, it is critical to understand how this heterogeneity emerges during establishment of symbiosis, which drives the evolution, ecology, and physiology of both host and microbe.

While the ecological processes that create heterogeneity during community assembly have been studied with mathematical models (e.g., [16]), validation of these models in empirical studies using natural, ecologically realistic communities, including host-associated microbial communities [12–14,17], is scarce. Some of these processes are deterministic, acting on specific traits that allow or hinder establishment of a taxon in a predictable, niche-based fashion. However, community assembly is also governed by dispersal between habitats. Dispersal imparts a stochastic element on community assembly [14]: Taxa immigrate and establish in new patches in a probabilistic manner, in part because they experience transient reductions, called bottlenecks, in population size [18]. These bottlenecks intensify ecological drift (i.e., stochastic variation in community composition). Since the proposal of Hubbell's unified neutral theory of biodiversity, the relative role of stochastic processes such as ecological drift in community assembly, compared with deterministic niche-based processes such as between-species interactions, has been a matter of continuous study [19–21].

In the context of host–microbe mutualistic symbioses, hosts impose stringent ecological selection during community assembly by filtering out or sanctioning non-beneficial and pathogenic microbes [10,22–25]. While this paradigm can explain the consistency with which hosts can acquire symbionts while excluding nonsymbiotic taxa (Fig 1D), it does not explain how these symbiont communities differ between individuals (Fig 1E), nor can it account for spatial structure in communities within the host. To illustrate the importance of ecological drift during the establishment of even highly specific symbioses, we employ the squash bug, *Anasa tristis* (Fig 1A), as a model. *A. tristis* is host to specific symbionts in the β-proteobacterial genus *Caballeronia* (previously referred to in the literature as the *Burkholderia* "SBE" clade [26] or the *B. glathei*-like clade [27]), which it requires for survival and normal development to adulthood [28]. Acquisition of *Caballeronia* occurs through the environment after nymphs (immature insects) molt into the second instar. Once they successfully colonize the host, symbionts are housed in hundreds of sacs called crypts, which form 2 rows running along a specific section of the midgut, called the M4 (Fig 1B). Unlike many other insect symbionts, *Caballeronia* can be isolated from bugs and established in pure culture in the laboratory. Because *A. tristis* nymphs hatch from their eggs symbiont-free, the symbiosis can be reconstituted anew every generation by feeding cultured symbionts to these nymphs in the laboratory [7,29].

Because the host relies on *Caballeronia* strains as its symbiotic partners, its digestive tract imposes strong selection to favor *Caballeronia* colonization [25,30], as in other specialized systems [22]. As a result, *Caballeronia* constitutes the vast majority of the microbial community within the M4 symbiotic organ, even though squash bug nymphs are exposed to diverse environmental microbes on squash fruit and plants [28]. However, this and similar bug-*Caballeronia* symbioses are extremely nonspecific below the genus level [31], with distantly related symbiont isolates conferring nearly the same degree of host benefit [7,32]. In accordance with this apparent lack of specificity, we observe that within-host *Caballeronia* communities from wild squash bug populations vary widely in their composition [7] (Fig 1C). So, beyond the coarse ecological filter that the host insect applies against nonsymbiotic taxa [25,30], little is

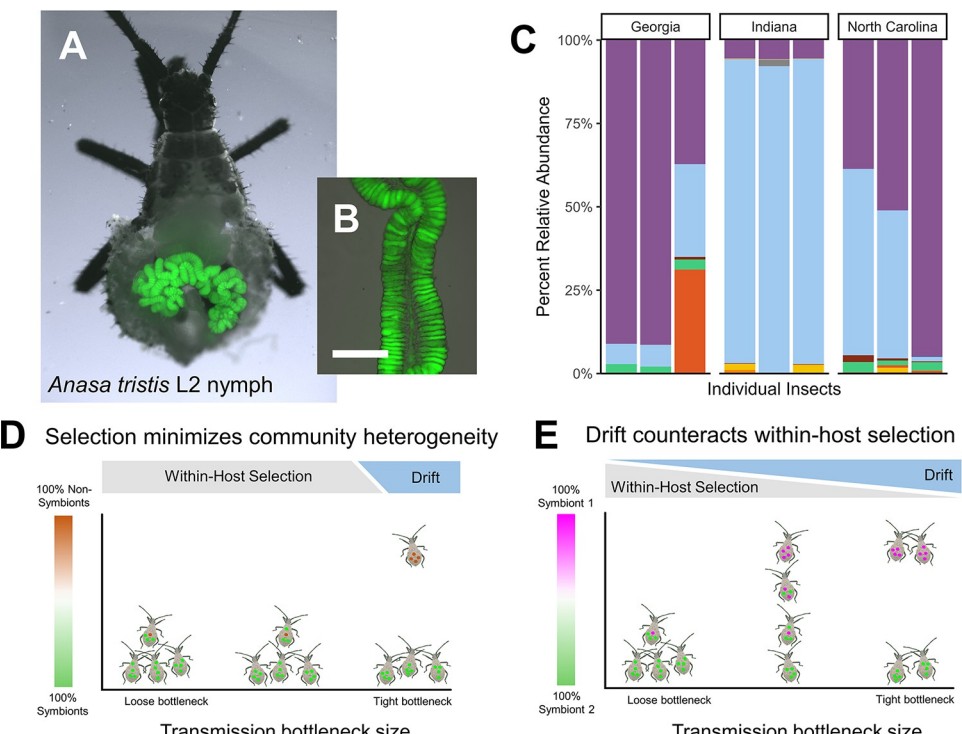

**Fig 1. The squash bug, *A. tristis*, engages in a specialized host–microbe symbiosis with bacteria in the genus *Caballeronia*.** (A) A second instar squash bug (*A. tristis*) dissected to reveal the M4 section of the midgut, colonized with *Caballeronia* symbionts expressing the sfGFP fluorescent protein. (B) Inset: The fine structure of the M4, consisting of 2 rows of hundreds of sac-like crypts lining a central lumen. Crypts are colonized at high density with *Caballeronia* symbionts expressing the sfGFP fluorescent protein. The scale bar represents 250 μm. (C) Relative abundance of the top 20 *Caballeronia* 16s V3-V4 amplicon sequence variants (ASVs) within 9 bugs across 3 field localities in Georgia, Indiana, and North Carolina, with different colors representing different ASVs. The data underlying this figure can be found in S1 Data. (D) A common notion is that because hosts with obligate microbial symbionts are under uniquely strong selection to transmit and maintain particular symbiotic taxa, selection (gray) rather than drift (blue) broadly governs the outcome of community assembly in these highly specialized microbiomes. In this paradigm, communities (left and middle clusters) invariably are dominated by symbionts (green) that have higher within-host fitness than non-symbionts (brown). As a result, rare, highly divergent communities (top right) containing non-symbionts might emerge only by chance, when symbionts undergo extremely tight transmission bottlenecks (right cluster). However, this paradigm does not explain compositional heterogeneity among symbiont communities, in which competing symbiont strains may be functionally identical in host benefit and ability to colonize. (E) We posit that ecological drift plays an important role in structuring symbiont strain diversity by counteracting within-host selection in symbiont populations during colonization. As in (D), selection (gray) acts on differences in relative fitness between competing strains (magenta and green) to drive community convergence towards a mean composition (left cluster). However, the effect of selection is only apparent when symbiont populations undergo weak population bottlenecks during host colonization. Drift (blue) minimizes the impact of selection, such that even if competing symbiont strains differ substantially in within-host fitness, heterogeneity in community composition emerges as if this fitness difference is absent, i.e., neutrally (middle and right clusters).

known about the ecological processes that maintain within- and between-host diversity of this beneficial symbiont.

Here, we explore the hypothesis that both within- and between-host diversity in symbiont populations arise stochastically as a result of ecological drift during infection [14]. First, we set out to explore a range of conditions under which this pattern might emerge, incorporating neutral competition (where all cells are isogenic, and thus functionally equivalent, individuals) [20] and interspecies competition (where cells are genetically distinct, but still equally host beneficial) between symbiont strains. By experimentally manipulating transmission bottleneck size, we show that ecological drift alone can account for heterogeneity between hosts,

segregating strains between hosts and decreasing the probability of coinfection. Using isogenic coinfections, we additionally demonstrate that the symbiotic organ imposes spatial heterogeneity on within-host populations, whereby separate crypts are colonized by different strains. Our results demonstrate the role of ecological drift in the assembly of a highly specialized host–microbe system and in structuring symbiont population diversity across scales.

## Results

### Ecological drift is sufficient to generate variation in colonization outcome

We reasoned that if ecological drift plays a role in generating heterogeneity in symbiont populations between hosts, it should generate greater and greater heterogeneity under smaller and smaller inoculum densities, which represent tightening transmission bottlenecks in our experiments. Specifically, the neutral model [33] implies that under tight bottlenecks, which shrink the effective size of a local community, colonization outcomes should be bimodal, with hosts dominated by clonal lineages, regardless of strain identity [16,19]. By contrast, when host control determines colonization, altering the inoculum size should have minimal impact on community composition across hosts. Additionally, if strong competition between symbionts determines the outcomes of colonization, individual hosts should be mono-colonized across a broad range of inoculum densities. To test this, we implemented a simple experimental design (Fig 2A), previously applied to human pathogens and legume nodule symbionts, that modulates transmission bottleneck size while maintaining the relative abundance of each strain during transmission [34,35]. To minimize the involvement of selection, we used isogenic, green- and red-fluorescently labeled isolates of *C. zhejiangensis* GA-OX1, a highly beneficial strain isolated previously from *A. tristis* [7]. Because our experiments involved only 2 competitors, we used the bimodality coefficient [14,36] to quantify heterogeneity in community composition. The bimodality coefficient is a composite measure of skewness and kurtosis, which is maximized for a distribution with equal weight at the extrema of the data. As this measure can be sensitive to small sample sizes, we also calculated the Hartigan's dip statistic, which represents a more robust general measure of multimodality. We inoculated second instar squash bug nymphs with approximately 1:1 mixtures of GA-OX1 sfGFP with GA-OX1 RFP, diluted to produce inocula ranging from approximately $10^6$ to $10^1$ CFU/μL (S1A Fig). These inoculum densities are within the natural range of variation in symbiont density that hosts might encounter, whether in freshly deposited adult feces ($10^5$ to $5 \times 10^6$ CFUs/μL), on which nymphs can feed to acquire symbionts [37], or in soil, in which *Caballeronia* is present at lower densities alongside many other microbes [38,39].

Consistent with the neutral model, under the highest inoculum densities, corresponding to the loosest bottlenecks, differences between the M4 communities of individuals are minimized, with a slight bias in favor of the sfGFP strain (Figs 2B and S1B). The slight bias towards sfGFP colonization could be due to toxic aggregation of the dTomato fluorescent protein, which has been observed in eukaryotic cells [40]. As inoculum density decreases, and thus as transmission bottlenecks tighten, individual infections become increasingly dominated by one or the other strain, causing the bimodality coefficient to increase (Figs 2C and S1C and S1 Table). Below 100 CFU/μL, individual infections are comprised of mostly either sfGFP or RFP, manifesting as a weakly but significantly bimodal outcome (bimodality coefficient = 0.677, Hartigan's dip statistic = 0.152, $p < 2.2 \times 10^{-16}$) (S1 Table). Fluorescence images of whole nymphs provided qualitative confirmation of our results, with more heterogeneity observed between nymphs at lower inoculum densities (S2 Fig). Through this set of experiments, we show that ecological drift is sufficient to drive heterogeneous colonization outcomes.

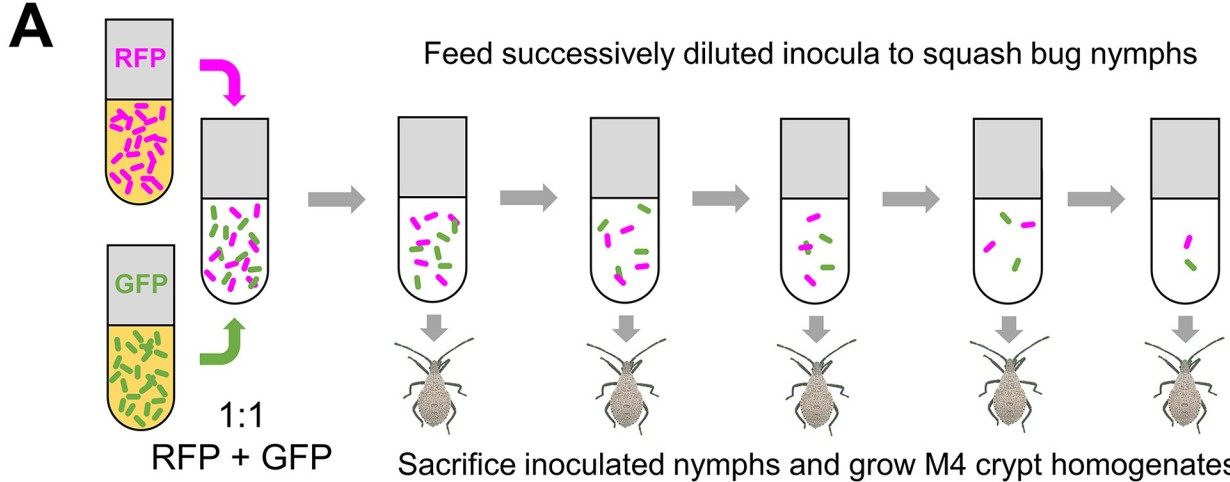

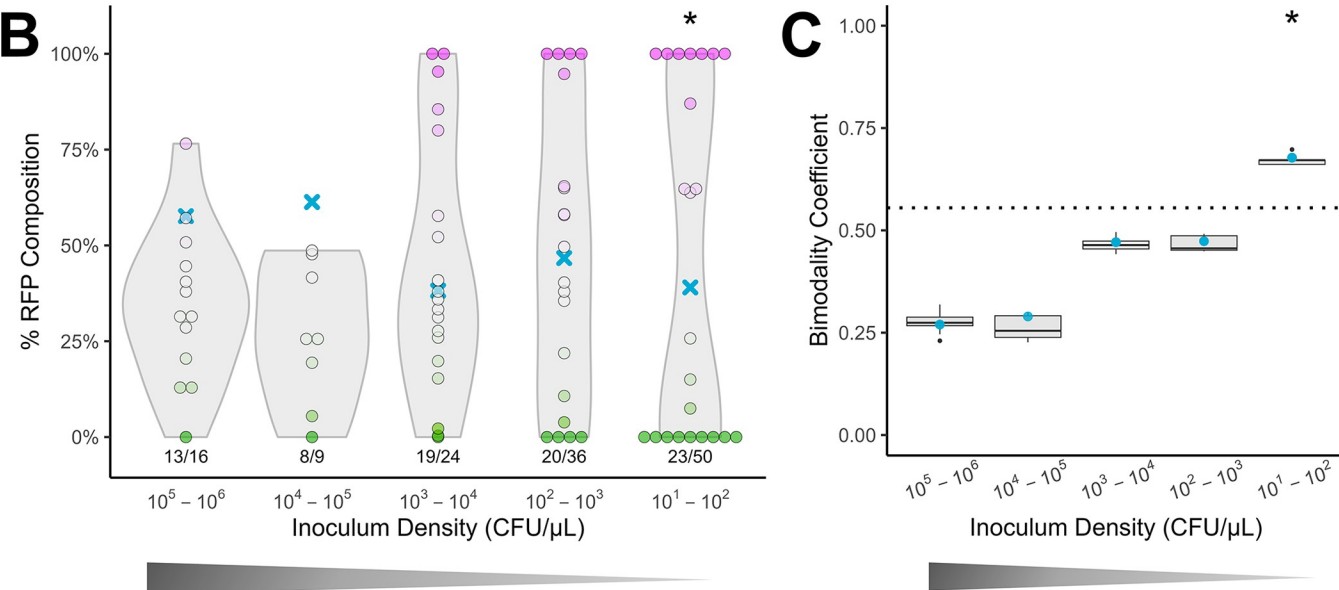

**Fig 2. The strength of ecological drift mediates variability in the outcome of symbiont colonization.** (A) Experimental design. Symbionts previously isolated from squash bugs, made to stably express green or red fluorescent proteins (GFP or RFP), are grown individually in liquid culture. Liquid cultures are combined at a predetermined ratio, then the mixture is diluted at various concentrations in the inoculation medium such that the inoculum density (a proxy for transmission bottleneck size) varies over several orders of magnitude while retaining the relative abundance of each strain across inocula. (B) Variable colonization outcomes associated with different transmission bottleneck sizes in isogenic co-inoculation, using *Caballeronia zhejiangensis* GA-OX1 sfGFP and RFP. Blue X marks indicate the mean % GA-OX1 RFP associated with each inoculum treatment, ranging from $10^1$ to $10^6$ CFU/μL. Points represent individual nymphs, and the color of each point and its position along the y-axis represent the percent relative abundance of GA-OX1 RFP colonies among all fluorescent colonies recovered from each nymph. Magenta points represent nymphs from which only RFP colonies were recovered, green points represent nymphs from which only sfGFP colonies were recovered, and faded magenta/green colonies represent coinfected nymphs. Violin plots associated with each treatment depict the shape of the distribution in relative RFP abundance. Asterisks indicate significantly multimodal infection outcomes as determined by Hartigan's dip test, at a significance value of $p < 0.05$. Below each violin plot, the success rate of colonization is indicated, as the number of nymphs that were successfully colonized with *Caballeronia* out of all nymphs sampled. Trials were aggregated across multiple runs. The data underlying this figure can be found in S2 Data. (C) Bimodality coefficients calculated from results in panel B. Large blue dots indicate bimodality coefficients calculated from all bugs in each treatment; boxplots indicate bimodality coefficients calculated by jackknife resampling in each treatment. The 0.555 threshold (marked with a dotted line) indicates the bimodality coefficient expected from a uniform distribution. Asterisks indicate significantly multimodal infection outcomes as indicated by Hartigan's dip test, at a significance level of $p < 0.05$. The data underlying this figure can be found in S2 Data.

## Ecological drift maintains coexistence between competing strains across separate hosts

Having illustrated the action of transmission bottlenecks on a single symbiont genetic background, we next sought to understand how they would act on genetically distinct host-beneficial strains. If ecological drift has an effect even when selection can act on competitive differences between strains, we should see a similar result to our previous experiment, with bimodality increasing with tightening transmission bottlenecks. We tested *C. zhejiangensis* GA-OX1 alongside *C. sp. nr. concitans* SQ4a [28], which represent 2 lineages within the *Caballeronia* genus (S3 Fig) [27,41] but are nonetheless equally beneficial for host developmental time and survivorship in the laboratory [7,28]. SQ4a was previously labeled with GFPmut3 [28,42] and was additionally labeled with sfGFP and dTomato for this study using the same constructs [43] that were applied to GA-OX1 above.

First, we demonstrated that GA-OX1 and SQ4a compete under an in vitro approximation of natural conditions within the host midgut (Fig 3A). In trials where SQ4a sfGFP and RFP were grown together as liquid cultures in filter-sterilized zucchini squash extract, both strains were recovered at high densities after 24 h. On the other hand, when either SQ4a strain was grown with a counter-labeled GA-OX1, SQ4a almost always went extinct (*T* test, $p < 0.001$, $n = 10$). Labeled GA-OX1 strains grew to high densities regardless of whether they were growing alongside SQ4a or the counter-labeled GA-OX1. These data suggest that GA-OX1 is the superior competitor to SQ4a under these culture conditions.

We next co-colonized hosts with mixtures of GA-OX1 RFP and SQ4a GFPmut3, using the same experimental design as in isogenic-strain colonization. Moderately high to low inoculum densities all resulted in strong bimodality in infection outcomes, where individual hosts were dominated either by GA-OX1 or by the competitor SQ4a (Fig 3B and 3C and S2 Table). Only at high inoculum density ($\geq 10^4$ CFUs) were infections biased in favor of GA-OX1 (Fig 3C). This bimodality was qualitatively reproducible across different combinations of SQ4a and GA-OX1 expressing different fluorophores from different synthetic constructs (Figs 3B and S4 and S3 and S4 Tables). Bimodality coefficients were consistently higher at high colonization densities in interspecific competition experiments than neutral competition experiments (Figs 3D, S4B, and S4D and S2–S4 Tables); this is expected, as neutral competition should produce unimodal populations when colonization rates are high.

## Ecological drift during colonization generates within-host spatial heterogeneity

The squash bug symbiotic organ, called the M4, contains hundreds of crypts (Fig 1A and 1B). Because each crypt is filled with its own population of symbionts, we asked whether symbiont composition might exhibit between-crypt heterogeneity within the host, consistent with previous unquantified observations from related insect-*Caballeronia* models [25,44]. If crypts indeed contain heterogeneous populations, we would expect crypts to contain mostly RFP- and mostly GFP-expressing symbionts, as opposed to highly similar populations composed of one or both types. We also asked if within-host heterogeneity might be sensitive to inoculum density in the same manner as between-host heterogeneity. If so, we would expect greater heterogeneity among crypts within a host when symbionts are subjected to tighter transmission bottlenecks during host colonization.

We systematically characterized within-host spatial heterogeneity by co-inoculating nymphs with 1:1 mixtures of counter-labeled GA-OX1 at approximately $10^6$ and $10^2$ CFU/μL, as above. Co-infected nymphs were selected by screening whole insects under fluorescence prior to dissection. By imaging freshly dissected whole guts from coinfected nymphs, we

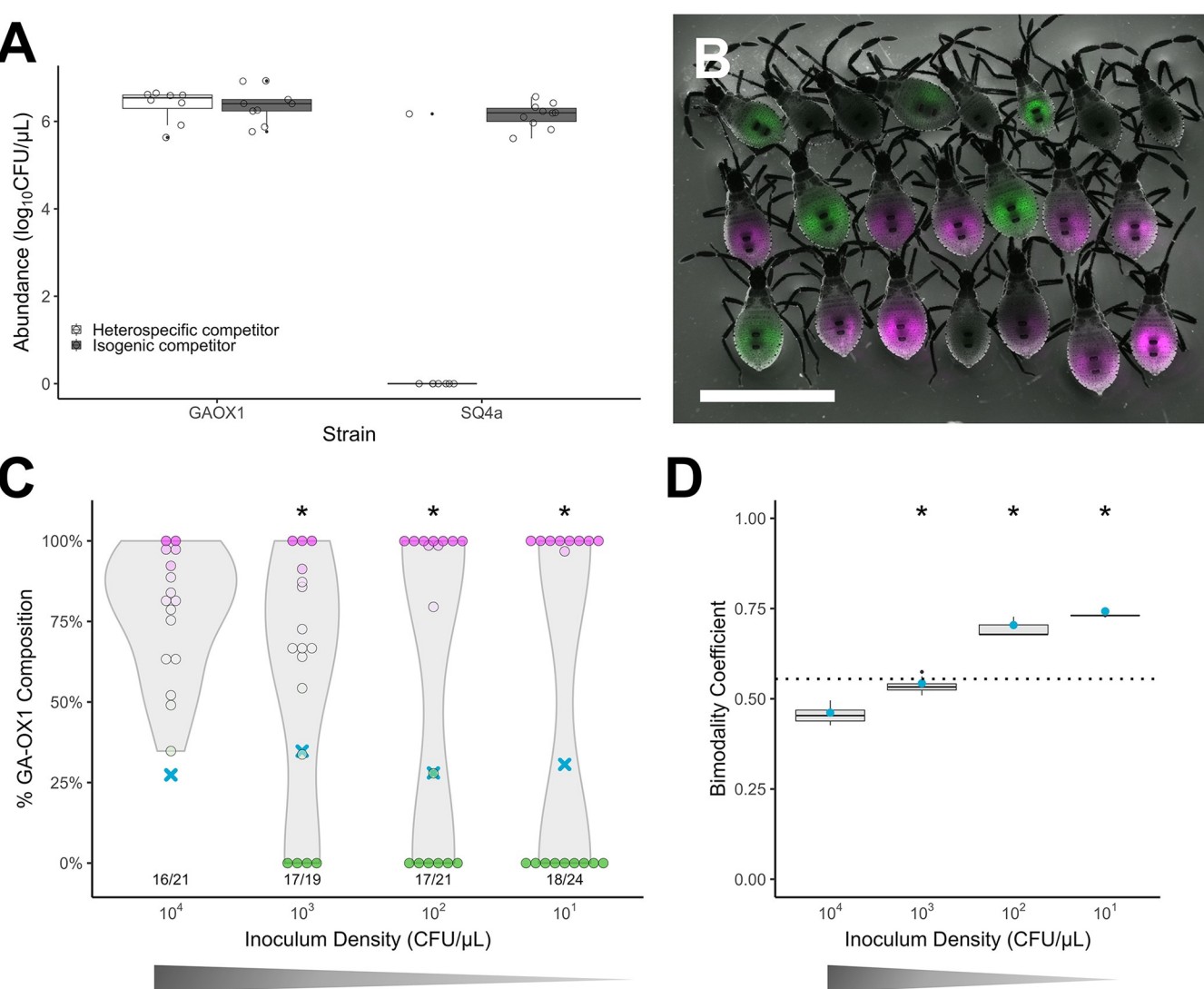

**Fig 3. Ecological drift mediates heterogeneity in host infection by competing strains.** (A) Competitive interactions between 2 symbiont strains in filter-sterilized zucchini squash extract. GA-OX1 and SQ4a differentially labeled with sfGFP or RFP were cocultured with a strain with the opposite fluorescent marker, either of an isogenic background or of the other species, and then assayed after 24 h for abundance. Data are pooled across fluorophore reciprocal swaps. White boxplots represent growth with a heterospecific competitor, while gray boxplots represent growth with an isogenic competitor. The data underlying this figure can be found in S3 Data. (B) Fluorescence image of one cohort of squash bug nymphs fed a combination of SQ4a sfGFP and GA-OX1 RFP, at a combined density of approximately 5,000 CFU/μL. Green indicates the presence of SQ4a sfGFP in nymphs, and magenta indicates the presence of GA-OX1 RFP. Nymphs without fluorescence were not successfully colonized with either symbiont strain. Scale bar indicates 5 mm. (C) Variable colonization outcomes associated with different transmission bottleneck sizes in two-species co-inoculation, using *C. sp. nr. concitans* SQ4a GFPmut3 and *C. zhejiangensis* GA-OX1 RFP. Blue X marks indicate the percent GA-OX1 RFP associated with each inoculum treatment, ranging from $10^1$ to $10^4$ CFU/μL. Points represent individual nymphs, and the color of each point and its position along the y-axis represent the percent relative abundance of GA-OX1 RFP colonies among all fluorescent colonies recovered from each nymph. Magenta points represent nymphs from which only GA-OX1 RFP colonies were recovered, green points represent nymphs from which only SQ4a GFPmut3 colonies were recovered, and faded magenta/green points represent coinfected nymphs. Violin plots associated with each treatment depict the shape of the distribution in relative GA-OX1 RFP abundance. Below each violin plot, the success rate of colonization is indicated, as the number of nymphs that were successfully colonized with *Caballeronia* out of all nymphs sampled. Asterisks indicate significantly multimodal infection outcomes as determined by Hartigan's dip test, at a significance level of $p < 0.05$. The data underlying this figure can be found in S4 Data. (D) Bimodality coefficients calculated from results in panel C. Large blue dots indicate bimodality coefficients calculated from all bugs in each treatment; boxplots indicate bimodality coefficients calculated by jackknife resampling in each treatment. The 0.555 threshold (marked with a dotted line) indicates the bimodality coefficient associated with a uniform distribution. Asterisks indicate significantly multimodal infection outcomes as indicated by Hartigan's dip test, at a significance value of $p < 0.05$. The data underlying this figure can be found in S4 Data.

observed that the M4 does impose spatial heterogeneity on symbiont populations, with individual crypts varying in GFP and RFP intensity even at colonization with $10^6$ symbiont CFU/μL (Fig 4A). However, there is a clear gradient in the degree of heterogeneity among crypts along the length of the M4, with anterior crypts being co-colonized and posterior crypts being

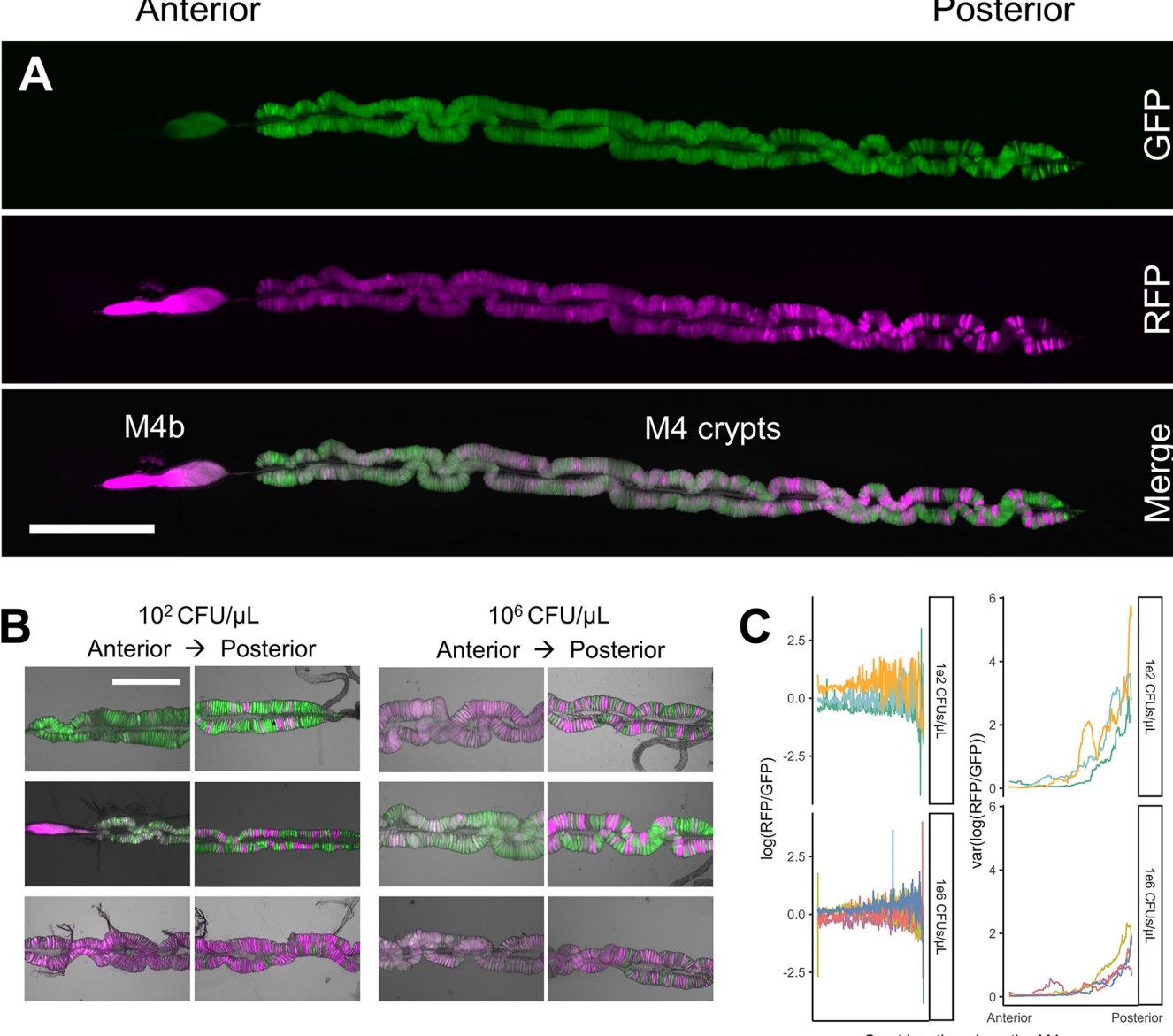

**Fig 4. The squash bug symbiotic organ (the M4) imposes spatial heterogeneity on *Caballeronia* populations within the host.** (A) Tilescan of the entire M4 of a representative second-instar nymph fed combined $10^6$ CFU/μL GA-OX1 sfGFP and RFP, dissected and linearized to illustrate symbiont colonization along its length. Individual panels represent the merged GFP and RFP channels (top), only the GFP channel (middle), and only the RFP channel (bottom). In each panel, the anterior end of the M4 is oriented to the left and the posterior end is oriented to the right. The intensely magenta, spindle-shaped organ is the M4b, which is functionally distinct from the crypts that house the symbiont population. The scale bar represents 1 mm. (B) Anterior and posterior crypts from 3 nymphs fed $10^2$ CFU/μL (left) and 3 other nymphs fed $10^6$ CFU/μL (right) GA-OX1 sfGFP and RFP, dissected and prepared as in (A). Panels represent the merging of GFP, RFP, and DIC (differential interference contrast) channels. For each specimen, the anterior crypts are on the left and the posterior crypts are on the right. The scale bar represents 500 μm. Raw images for all specimens are available at https://doi.org/10.15139/S3/YZPBGY. (C) Ratio of normalized RFP intensity relative to normalized GFP intensity (left) and variance in this ratio within a sliding window (right) along a transect from the anterior to the posterior of the M4. Nymphs were either inoculated with $10^2$ CFU/μL ($n$ = 3, top) or $10^6$ CFU/μL ($n$ = 4, bottom), and different colored lines represent the trajectories of these values associated with each nymph. The data underlying this figure can be found in S7 Data.

singly infected (Fig 4B). We quantified this gradient by measuring the variance in RFP intensity relative to GFP along the length of the M4 (Fig 4C). Contrary to our expectations, we saw that nymphs colonized with just $10^2$ symbiont CFU/μL also exhibited this gradient, with anterior crypts being co-colonized despite a 10,000-fold reduction in inoculum density (Fig 4B and 4C). Thus, patterns of heterogeneity within the host are consistent over 4 orders of magnitude in inoculum density. Even when microbe–microbe competition is nearly neutral, host anatomy appears to impose spatial structure on symbiont populations.

## Discussion

Previous research on a suite of closely related insect-*Caballeronia* symbioses has demonstrated both heterogeneity in symbiont composition and low diversity of symbiont populations within hosts [7,31,45,46]. In the present study, we reveal the processes that underlie these patterns are consistent with stochastic colonization, which results in strong ecological drift as symbionts establish in their host insects. By modulating transmission bottleneck sizes of inocula containing isogenic, nearly neutrally competing strains, we show that ecological drift alone can generate heterogeneity in colonization outcome between different hosts, consistent with the neutral theory of biodiversity [20,21]. The transmission bottlenecks in our experiments are likely to be within the range of natural variation in transmission bottleneck size through natural routes [37,38] of symbiont transmission in *A. tristis* and related insects [47], suggesting that our results pertain to how drift affects symbiont population structure in wild bug-*Caballeronia* assemblages.

Next, by manipulating bottleneck sizes of inocula containing different symbiont species, we not only experimentally demonstrate the role of ecological drift in maintaining genetic diversity but also highlight the role of between-symbiont competition. Ecological drift generates variation in founding populations between hosts, while competition drives homogeneity in symbiont populations during subsequent proliferation inside a single host. The effect is bimodality in symbiont colonization, even when transmission bottlenecks are loose. Our results mirror findings from similar studies using plant communities, where competitive asymmetries between species also exaggerate the effect of ecological drift [19], and call attention to the role that inter-symbiont competition might play during the early stages of colonization in other host–microbe systems [13,48–50].

### Generality of drift in generating heterogeneity in symbiotic systems

Although we implemented our experiments using horizontal transmission, many symbioses exhibit elaborate, host-controlled mechanisms that ensure vertical transmission [51–55]. Despite host control, vertically transmitted symbionts, including obligate insect mutualists [56], are not immune to the effects of ecological drift, which acts on communities regardless of how they disperse. Indeed, some vertically transmitted symbionts undergo extreme transmission bottlenecks [57–59], exaggerating the intensity of drift, and vertically transmitted symbionts also compete for host colonization [59–62]. Thus, we should expect drift to generate heterogeneity in vertically transmitted infections [63,64] in a similar manner as we observed in our horizontal transmission experiments.

Based on our findings, we argue that the role of ecological drift is inadequately considered in host–microbe associations [13,14,34,35,65]. Notably, as long as ecologically overlapping microbes are capable of colonizing a within-host niche, host benefit, partner choice, and coevolutionary history may be unnecessary to explain between-host variation in microbiomes [2,5,15,49,64,66–69]. This is of course not to say that niche-based ecology does not affect these communities or their evolution (e.g., [70]). In our system, there is limited diversity in the

microbial symbiont community, and interactions between strains are likely dominated by competition [5,44,59,71,72]. By contrast, multispecies communities may contain facilitative interactions, such as signaling crosstalk, cross-feeding, metabolic division of labor, and complementary host-provisioning [8,73–80], all of which could fundamentally alter ecological dynamics. Nonetheless, the neutral model often performs surprisingly well in explaining patterns in multispecies communities [21], suggesting that drift and other stochastic processes should at least be considered when attempting to explain patterns of diversity within more specialized symbioses as well.

The pervasive effect of ecological drift suggests it may also play a key but undervalued role in the evolution of specialized host–microbe symbioses. First, ecological drift can override selection and maintain strain variation within a host population, by providing refugia for suboptimal or less competitive symbionts. In addition, by driving compositional variation between host-associated microbiomes, ecological drift can expose taxonomically or functionally distinctive strains and communities to selection [61,81–85]. If a distinctive microbiome can maintain its association with a particular host lineage, coevolution with the host may eventually occur. By simultaneously maintaining genetic variation among symbionts and generating heterogeneity in symbiont community composition, we argue that ecological drift could provide another explanation for the paradox of variation in host–microbe mutualisms [9].

Beyond its role in generating between-host heterogeneity, ecological drift also generates heterogeneity in symbiont populations within a host. In the squash bug, we found that gut crypts, a unique anatomical feature of the M4 symbiotic organ, generate heterogeneity by segregating strains into discrete compartments within the same host. This has parallels in other symbiotic organs, including the crypts in the light organs and accessory nidamental glands of sepiolid squid [86–88], coralloid roots and root nodules in cycads and legumes [35,89,90], and pores in human skin [91]. Because we show that such compartmentalization acts even on isogenic cells, we propose that within-host population spatial structure, as with between-host population structure, is not adequately explained by either host selection or microbial competition, and is instead characterized by stochastic colonization of different crypts [35]. While spatial heterogeneity frequently emerges as a result of between-strain interactions within in vitro communities [92–96], here, the anatomy of a host forcefully imposes it even in the apparent absence of such interactions. How squash bugs and other multicellular hosts benefit from subjecting their symbiont populations to such elaborate compartmentalization remains an open question [97,98].

Although we have discussed how ecological drift results in segregation of genetic variation within the host, we were surprised to find that population diversity within individual crypts is apparently independent of inoculum density. We expected that the degree of population admixture within the crypts would depend on inoculum density, with crypts being predominantly coinfected at high inoculum density and predominantly singly infected at low inoculum density, as has been demonstrated in vitro systems on plates and in microfluidics experiments [99,100]. However, we instead observed an anterior-posterior gradient of admixture for all co-colonized bugs, consistent across 4 orders of magnitude in initial inoculum density. This suggests that in vivo colonization processes impose distinct conditions that generate structure in *Caballeronia* populations. We know almost nothing about symbiont colonization at the single-cell level in the squash bug. However, we speculate that the host permits colonization of individual crypts by only a limited number of symbiont cells and that inoculation of individual crypts continues to some extent after the initial colonization event by movement of propagules within the symbiont organ. Further study is necessary to ascertain whether coinfection within single crypts affects within-host symbiont evolution and host fitness, as predicted by others [101], or creates opportunities for horizontal gene transfer [102].

In this work, we illustrate the role of ecological drift in shaping symbiont host populations at multiple scales. Our findings highlight the effect of ecological drift during colonization by maintaining heterogeneity in symbiont populations both within and between hosts. We posit that ecological drift can weaken selection from microbe–microbe interactions within the host, while also setting the stage for the evolution of these same processes. These results contribute to our understanding of the role that stochastic dynamics play in the assembly of ecological communities, even in ancient, highly specific host–microbe associations subjected to extensive host control [21].

## Methods

### Study system

Squash bugs (*Anasa tristis*) were maintained on yellow crookneck squash plants (*Cucurbita pepo* "Goldstar") in 1 ft³ mesh cages. Hatchlings were maintained on pieces of surface-sterilized organic zucchini in plastic rearing boxes, where they remain aposymbiotic (i.e., *Caballeronia*-free, though not necessarily free of other microbes). Hatchlings molt to the second instar, the life stage competent for symbiont colonization, after 2 days of feeding. Nymphs utilized in this experiment were typically 1 week old or less.

*Caballeronia* symbionts *C. sp*. SQ4a and *C. zhejiangensis* GA-OX1 were originally isolated from wild squash bugs at different localities in northeastern Georgia, United States of America. SQ4a and GA-OX1 form phenotypically very distinct colonies on nutrient agar (NA; 3 g/L yeast extract, 5 g/L peptone, 15 g/L agar) and are not closely related within the genus *Caballeronia* [41] (S3 Fig). Cultures were typically grown on NA plates or in Luria Bertani (LB) Lennox broth with low salt (Sigma-Aldrich L3022) at 25˚C. Unless otherwise stated, 2 ml broth cultures were initiated from colony picks of 3- to 4-day-old colonies grown on NA at 25˚C, and grown overnight with shaking at 200 rpm at 25˚C.

### Strain construction

The mini-Tn7 system [103] facilitates the stable, orientation-specific introduction of foreign DNA into bacterial genomes at a neutral intergenic site, *attTn7*, with minimal effects on phenotype and fitness in vitro [104–106]. To make readily distinguishable but otherwise isogenic symbiont strains, we genomically integrated a green fluorescent protein (sfGFP; henceforth GFP) and a red fluorescent protein (dTomato; henceforth RFP) into SQ4a and GA-OX1 using improved versions of previously developed mini-Tn7 vectors (Table 1) [103]. The conjugative *Escherichia coli* K12 strain SM10(λpir) harboring pTn7xKS-sfGFP or pTn7xKS-dTomato (Table 1), which were a generous gift from Travis Wiles [43], as well as an *E. coli* parent of the same strain harboring helper plasmid pTNS2 [103], were plated with SQ4a and GA-OX1 at high density on LB plates with salt (10 g/L tryptone, 5 g/L yeast extract, 10 g/L NaCl, 15 g/L agar). After 24 to 48 h of incubation at 30˚C, matings were harvested into LB Lennox low-salt broth with a lytic coliphage, T7, to eradicate *E. coli*. After further incubation for 4 h at 30˚C shaking at 200 to 225 rpm, cultures were plated on NA amended with 1 mM isopropyl-β-D-1-thiogalactoside (IPTG) and 10 µg/ml gentamicin to select for successful integrants. Colonies on selective plates were screened for fluorescence and frozen at −80˚C as 20% v/v glycerol stocks.

To confirm stability of fluorophore expression, newly constructed strains were streaked on NA plates and visually assessed for fluorescence after 2 days. To confirm site- and orientation-specificity of mini-Tn7-GmR integration, we ran PCR to amplify the fragment between the endogenous *glmS* gene (GA-OX1: 5′ AGGCGCGTTGAAGCTCAAGG 3′; SQ4a: 5′ CGCTGGAGCCGCAAATCATC 3′) and the inserted *aacC* gentamicin resistance marker

**Table 1. Strains and plasmids used in this study.**

| Strains | Description | Source/reference |
|---|---|---|
| *Escherichia coli* SM10(λpir) | recA::RP4-2-TcR::Mu, pir+ conjugative strain | Travis Wiles [107,108] |
| *Caballeronia zhejiangensis* GA-OX1 | Wild *Anasa tristis* isolate, Oxford College Organic Farm | [7] |
| *Caballeronia zhejiangensis* GA-OX1 sfGFP | Mini-Tn7T-GmR-sfGFP derivative of GA-OX1; high fluorophore expression driven by a tac promoter | [37] |
| *Caballeronia zhejiangensis* GA-OX1 RFP | Mini-Tn7T-GmR-dTomato derivative of GA-OX1; high fluorophore expression driven by a tac promoter | [37] |
| *Caballeronia sp. nr. concitans* SQ4a | Isolated from a wild-caught *Anasa tristis* held in captivity; originally collected at Oakhurst Community Garden | [28] |
| *Caballeronia sp. nr. concitans* SQ4a GFPmut3 | mini-Tn7-KmR-GFPmut3 derivative of SQ4a; low fluorophore expression driven by a $P_{A1/04/03}$ promoter | [28,109] |
| *Caballeronia sp. nr. concitans* SQ4a sfGFP | Mini-Tn7T-GmR-sfGFP derivative of SQ4a; high fluorophore expression driven by a tac promoter | [37] |
| *Caballeronia sp. nr. concitans* SQ4a RFP | Mini-Tn7T-GmR-dTomato derivative of SQ4a; high fluorophore expression driven by a tac promoter | [37] |
| Plasmids | Description | Source/reference |
| pTNS2 | R6K γ oriV, AmpR, tnsABCD | Travis Wiles [103] |
| pTn7xKS-sfGFP | colE1 oriV, mini-Tn7T-GmR-sfGFP, AmpR, lacI $P_{tac}$ hokB ghoT tisB | Travis Wiles [43] |
| pTn7xKS-dTomato | colE1 oriV, mini-Tn7T-GmR-dTomato, AmpR, lacI $P_{tac}$ hokB ghoT tisB | Travis Wiles [43] |

(aacC-83F: 5′ GTATGCGCTCACGCAACTGG 3′). We did not screen for insertion at additional sites, as to our knowledge the strains of *Caballeronia* we used have only 1 *attTn7* site.

## Competition assays in vitro

During the summer growing season, squash bugs feed on macerated cell contents, xylem, and phloem in tissues from squash plants and fruits [110,111]. To replicate microbial competition in this environment, competition assays in liquid culture were conducted in filter-sterilized zucchini squash extract. In short, juice from organic zucchini fruits was extracted in a juicer, combined, and filtered to remove large suspended particles. This filtrate was then centrifuged at 10,000 ×g for 3 h to pellet suspended particles, then filter-sterilized through a 0.2 μm filter and stored at −20˚C.

To initiate competition assays, GA-OX1 and SQ4a, labeled with sfGFP or dTomato as described above, were initially streaked from frozen 20% glycerol stocks onto NA plates and incubated at 30˚C for 48 h. Individual colonies from each plate were inoculated into 2 ml of LB media and incubated in a shaking incubator (New Brunswick Scientific Excella E25) at 25˚C for 12 h with shaking at 225 rpm. All cultures were equalized to an optical density (OD) of 1.0 by adding 100 μl of each culture to a 96-well plate and taking readings with a Synergy HTX multimode plate reader. The equalized cultures were spun down with an Eppendorf centrifuge 5424 R and washed with 1 ml of 1× phosphate-buffered saline (PBS) 3 times.

Monocultures of GA-OX1 and SQ4a were combined to form counterlabeled self versus self and self versus competitor cocultures, for a total of 4 combinations. Self versus self cocultures contained the same *Caballeronia* strain, differing only in the fluorescent protein, while self versus competitor cocultures contained different *Caballeronia* strains, also differing in the fluorescent protein. All cocultures were set up in 500 μl of a 1:1 mixture of filter-sterilized zucchini squash extract and PBS, and then incubated for 24 h at 30˚C with shaking at 225 rpm. As described above, cocultures were dilution plated on NA and incubated a further 20 to 24 h, and single colonies containing each fluorophore were distinguished and counted under a dissecting scope. We confirmed that SQ4a and GA-OX1 do not appear to inhibit each other

intensely on NA plates, suggesting that plating cocultures on NA provides an unbiased count for both competitors (S5 Fig).

## Competition assays in vivo

The generalized protocol for competition assays with varying transmission bottlenecks in vivo is presented in Fig 2A. GFP- and RFP-labeled *Caballeronia* strains were streaked out from glycerol stocks onto NA and incubated at 25°C for at least 3 days. To initiate liquid cultures, single colonies were picked into 2 ml LB in glass tubes and incubated at 25°C with shaking at 200 rpm; to account for different growth rates between SQ4a and GA-OX1, a glass inoculating loop was used to pick up an entire colony of SQ4a, while a p10 micropipettor tip was used to extract a small plug from part of a single GA-OX1 colony.

To prepare inocula for feeding, cultured bacterial cells were washed to remove LB. Two hundred μl of culture was spun down at 10,000 ×g at 4°C for 2 min. The supernatant was removed, and the pellet was resuspended in 1,000 μl 1× PBS. After a second centrifugation, the pellet was resuspended in 200 μl 1× PBS to bring the cells to their original culture density. For quality assurance, 10-fold serial dilutions were carried out using 30 μl washed cells in 270 μl PBS in a 96-well plate, and 50 μl each of the washed GFP- and RFP-labeled strains were then diluted into 400 μl of a complex feeding solution (a 1:1 mixture of filter-sterilized zucchini squash extract and PBS; for neutral competition trials) or a defined feeding solution (2% m/v glucose 10% v/v PBS; for interspecies competition trials), in each case containing 20 μl of a nontoxic blue dye (1 mM erioglaucine disodium). We found that the defined feeding solution improved nymphal feeding response and better prevented bacterial population growth during the inoculation time window, with minimal impact on our experimental results (S6 Fig). Nymphs previously starved overnight for 15 to 25 h in clean plastic rearing boxes (7 cm × 7 cm × 3 cm) were supplied with 120 μl of a single inoculum treatment blotted on quartered sectors of 55 mm diameter qualitative filter paper (Advantec MFS N015.5CM). Nymphs were then allowed to wander and feed ad libitum for 2 to 3 h. After this brief inoculation period, nymphs were housed singly in 24-well plates with small pieces of organic zucchini to develop for 3 days. Just before and after the inoculation period, inocula were serially diluted as above to quantify the concentration of each strain and ensure that no substantial growth or death of either strain occurred during the inoculation period (S6 Fig).

On the fourth day after inoculation, nymphs were killed in 70% denatured ethanol, surface sterilized in 10% bleach for 5 to 10 min, washed off again in 70% ethanol, and immersed in approximately 20 μl droplets of 1× PBS. Whole nymphs, when applicable, were imaged on an Olympus SZX16 stereomicroscope with an Olympus XM10 monochrome camera and Olympus cellSens Standard software ver. 1.13. Nymphs were immersed in a shallow volume of PBS in 6 cm plastic petri dishes, and images were taken in darkfield (30 ms exposure 11.4 dB gain), brightfield (autoexposure, 11.4 dB gain), a GFP channel (autoexposure, 18 dB gain), and an RFP channel (autoexposure, 11.4 dB gain). Darkfield and brightfield images were merged in FIJI version 1.54f using the Image Calculator plugin, and the result was then merged with the GFP channel, RFP channel, or both. M4s were individually dissected from nymphs, and the degree of green and red colonization was qualitatively estimated under a fluorescent microscope. Each M4 was then held in 300 μl 1× PBS in Eppendorf tube and crushed with a sterile micropestle; 30 μl of homogenate was serially diluted in 270 μl PBS and immediately dilution plated onto NA. Plates were then incubated at 30°C for about 24 h. Counts of GFP and RFP fluorescent colonies were recorded after refrigeration at 4°C for at least 24 h to enhance fluorescent protein expression. The count data of GFP and RFP colonies yielded by our sampling procedure almost always reflected our qualitative observations of GFP and RFP colonization

inside the M4, suggesting that our data accurately represent the colonization state within live insects.

## Microscopy of within-host symbiont populations

Bugs were inoculated and allowed to develop for 4 days as described above with approximately 60 and 931,000 CFU/μl inocula containing GA-OX1 GFP and RFP. We then intentionally screened individual insects for co-colonization, and only these insects were selected for dissection and microscopy. From each bug, the whole gut was dissected in a 20 to 30 μl droplet of PBS in a 30 mm diameter plastic dish. The M4 was stretched out to its full length, and straightened out as much as possible by severing tracheoles associated with the crypts and flipping the M4 over to minimize the number of twists in the M4. This was critical to minimize aberrations in fluorescence intensity and colocalization due to overlap between multiple crypts. The M4 was anchored at the posterior end by the tip of the bug abdomen and at the anterior end by the M1-M3 sections of the midgut, and cleaned several times by pipetting off debris, fat body, and hemocytes with clean PBS. Finally, the whole preparation was re-immersed in 2,550 μl of PBS, to which 1 μl of M9 buffer containing 1% Triton-X100 was added to aid the spreading of the droplet.

Gut preparations, which degrade or dry rapidly, were imaged as soon as possible. Tilescan images were taken using a Leica DMi8 inverted widefield light microscope with a Leica DFC9000 GT fluorescence camera and Leica Application Suite X ver. 3.4.2.18368 software. Automated tilescans were taken with a 10× objective lens with brightfield, DIC, GFP, and RFP channels. Fluorescent channels were established by filter sets. The GFP channel was set to: bandpass filter 470/40 nm emission, dichroic mirror 495 nm, emission 525/50 nm. The dsRed channel was set to: 546/11 nm excitation, dichroic mirror 560 nm, 630/75 nm emission. As each sample is unique, care was taken to set GFP and RFP channel exposure times manually according to the most intense pixels in the entire M4 (usually in the posterior crypts) to minimize signal saturation in any part of the preparation. Due to the convoluted shape of the M4, images were taken with and without autofocus, and stitched images were visually assessed to determine which images were more useful. The repetitive structure of the M4, composed of nearly identically sized, regularly spaced crypts, also necessitated a lower overlap value between tiles for tilescans, as low as 2%. LAS X software was used to merge tiles from tilescans without smoothing for quantitative analysis.

## Statistical analysis

All statistical analyses were conducted in R version 4.1.1, and the R package ggplot2 (version 3.4.2) was used for all data visualization. Because multiple trials were run for inoculation experiments, and some trials recovered very low numbers of infected nymphs, we binned nymphs from multiple trials into discrete treatment groups, based on inoculum size, for analysis. For neutral competition experiments, which utilized isogenic GA-OX1 GFP and RFP, we measured the proportion of GA-OX1 RFP extracted from each host. For interspecies competition experiments, utilizing different combinations of SQ4a and GA-OX1, we measured the proportion of GA-OX1 compared to the sum of all green and red fluorescent colonies extracted from each host.

Raw colony counts of each fluorescent strain recovered from each individual insect were converted into proportions for analysis and visualization. To quantify between-host heterogeneity in symbiont colonization for each inoculation treatment, we calculated a bimodality coefficient [36] using the R package mousetrap (version 3.2.0), as well as the population variance, for each inoculation treatment. Using the R package dip test (version 0.76–0), we also

implemented Hartigan's dip test [112], which calculates a dip statistic (S1–S4 Tables) based on the shape of the cumulative distribution function of a dataset. We considered a distribution to deviate from unimodality if *p*-values from the dip test repeatedly fell below the threshold of 0.05.

To quantify within-host spatial structure, a linear region of interest (ROI) was sampled from one complete row of crypts from each sample to obtain GFP and RFP intensities. Crossover of the ROI from one side of the M4 to the next was occasionally necessary to follow that row through each twist of the M4. The identical ROI was translated to obtain GFP and RFP intensities from the empty background immediately adjacent to the crypts. GFP and RFP intensities at each point along the M4 were normalized by subtracting the background signal from the same point outside the M4. For the RFP channel, the difference between crypt and background signal was occasionally less than 0; in these rare cases, the normalized RFP intensity at that point was assigned a value of 1. The log-transformed ratio of RFP to GFP intensity was measured from each pixel along the ROI. In addition, the variance in this value was calculated by iteratively sampling pixels from within a sliding interval 10% of the length of the ROI.

## Supporting information

**S1 Fig. Unaggregated infection outcomes resulting from neutral competition during isogenic co-colonization. The data underlying this figure can be found in S2 Data.** (A) Relative GA-OX1 RFP abundance within inocula used in isogenic colonization trials. Vertical lines demarcate which trials were aggregated for analysis in Fig 1. Blue X marks indicate the inoculum density, and the percent relative abundance of GA-OX1 RFP, in each trial. The dotted horizontal line represents the average relative abundance of GA-OX1 RFP (46.5%) across all trials. (B) Variable colonization outcomes associated with different transmission bottleneck sizes in isogenic co-inoculation, disaggregated from Fig 2B. Blue X marks indicate the inoculum density, and the percent relative abundance of GA-OX1 RFP, in each trial. Points indicate successfully colonized nymphs associated with each inoculation trial, and the color of each point and its position along the y-axis represent the percent relative abundance of GA-OX1 RFP colonies among all fluorescent colonies recovered from each nymph. Magenta points represent insects containing only RFP colonies, green points represent insects containing only GFP colonies, and faded magenta/green colonies are co-colonized. Note that multiple points overlap, particularly at the extremes of 0% and 100% RFP composition, due to the absence of jittering. (C) Bimodality coefficients calculated from unaggregated trials. Bimodality coefficients (black points) calculated from data in panel B. The 0.555 threshold (marked with a dotted line) indicates the bimodality coefficient expected from a uniform distribution.
(TIF)

**S2 Fig. Isogenic coinfections of *A. tristis* nymphs over 5 orders of magnitude in inoculum density.** Fluorescence images of nymphs from a single cohort colonized with different densities of GA-OX1 sfGFP and GA-OX1 RFP, ranging from $10^1$ to $10^6$ CFU/µl. Nymphs inoculated with only GA-OX1 sfGFP or only GA-OX1 RFP serve as controls (top 2 rows); the bottom 3 rows show nymphs from mixed inoculation trials (GA-OX1 GFP + RFP). Note that the red fluorescent protein dTomato is brighter in whole-body preparations of nymphs than the green fluorescent protein sfGFP, due to increased absorbance of green light by living tissue [113] and the high stability of free dTomato under physiological conditions.
(TIF)

**S3 Fig. Symbiont strains SQ4a and GA-OX1 represent distinct clades within the genus *Caballeronia*.** A whole genome-based phylogeny of selected species and previously isolated

*Anasa tristis* symbionts, representing major clades within the genus *Caballeronia* as defined by Peeters and colleagues [27], including the experimental strains *C. sp. nr. concitans* SQ4a and *C. zhejiangensis* GA-OX1. The phylogeny was constructed using RealPhy [114], with *Burkholderia cepacia* as the reference genome, using default settings except for a gap threshold of 0.1 and setting the model of evolution to GTR. Support values are bootstrap values based on 100 replicates. In addition to SQ4a and GA-OX1, *Caballeronia* strains A33M4c and IN-ML1 were also previously isolated from *A. tristis* [7,28]. SMT4a is a *Paraburkholderia terricola* soil isolate that can colonize *A. tristis* [28,41]. GenBank assemblies are as follows: GCF_023631065.1 (GA-OX1), GCF_022879815.1 (A33M4c), GCF_022627895.1 (*C. zhejiangensis*), GCF_023631085.1 (INML1), GCF_001544875.2 (*C. hypogeia*), GCF_000402035.1 (*C. insecticola*), GCF_023170545.1 (SQ4a), GCF_001544615.1 (*C. concitans*), GCF_902833485.1 (*C. glathei*), GCF_902859805.1 (*P. sediminicola*), GCF_022879555.1 (SMT4a), and GCA_009586235.1 (*B. cepacia*). Bootstrapping and tree files are available at https://doi.org/10.15139/S3/YZPBGY.
(TIF)

**S4 Fig. Different combinations of competing GA-OX1 and SQ4a fluorescent strains yield qualitatively similar responses to increasing stochasticity in transmission.** (A) Variable colonization outcomes associated with different transmission bottleneck sizes in two-species co-inoculation, using *C. sp. nr. concitans* SQ4a sfGFP and *C. zhejiangensis* GA-OX1 RFP. Blue X marks indicate the mean percent GA-OX1 RFP associated with each inoculum treatment, ranging from $10^0$ to $10^5$ CFU/μl. Points represent individual nymphs, and the color of each point and its position along the y-axis represent the percent relative abundance of GA-OX1 RFP colonies among all fluorescent colonies recovered from each nymph. Magenta points represent nymphs from which only GA-OX1 RFP colonies were recovered, green points represent nymphs from which only SQ4a sfGFP colonies were recovered, and faded magenta/green points represent coinfected nymphs. Violin plots associated with each treatment depict the shape of the distribution in relative GA-OX1 RFP abundance. Below each violin plot, the success rate of colonization is indicated, as the number of nymphs that were successfully colonized with *Caballeronia* out of all nymphs sampled. These values were not recorded for the $10^0$–$10^1$ treatment and thus omitted. Asterisks indicate significantly multimodal infection outcomes as determined by Hartigan's dip test, at a significance level of $p < 0.05$. The data underlying this figure can be found in S5 Data. (B) Bimodality coefficients calculated from results in panel A. Large blue dots indicate bimodality coefficients calculated from all bugs in each treatment; boxplots indicate coefficients calculated by jackknife resampling in each treatment. Colonization is bimodal (bimodality coefficient > 0.555) across several orders of magnitude of inoculum density. The 0.555 threshold (marked with a dotted line) indicates the bimodality coefficient associated with a uniform distribution. The data underlying this figure can be found in S5 Data. Note that for the $10^0$–$10^1$ treatment, the sample size was insufficient for jackknife resampling. (C) Variable colonization outcomes associated with different transmission bottleneck sizes in two-species co-inoculation, using *C. sp. nr. concitans* SQ4a RFP and *C. zhejiangensis* GA-OX1 sfGFP. Blue X marks indicate the mean percent GA-OX1 GFP associated with each inoculum treatment, ranging from $10^1$ to $10^5$ CFU/μl. Points represent individual nymphs, and the color of each point and its position along the y-axis represent the percent relative abundance of GA-OX1 GFP colonies among all fluorescent colonies recovered from each nymph. Magenta points represent nymphs from which only SQ4a RFP colonies were recovered, green points represent nymphs from which only GA-OX1 sfGFP colonies were recovered, and faded magenta/green points represent coinfected nymphs. Violin plots associated with each treatment depict the shape of the distribution in relative GA-OX1 GFP abundance.

Below each violin plot, the success rate of colonization is indicated, as the number of nymphs that were successfully colonized with *Caballeronia* out of all nymphs sampled. Asterisks indicate significantly multimodal infection outcomes as determined by Hartigan's dip test, at a significance level of $p < 0.05$. The data underlying this figure can be found in S6 Data. (D) Bimodality coefficients calculated from results in panel C. Large blue dots indicate bimodality coefficients calculated from all bugs in each treatment; boxplots indicate coefficients calculated by jackknife resampling in each treatment. Colonization is bimodal (bimodality coefficient >0.555) across several orders of magnitude of inoculum density. The 0.555 threshold (marked with a dotted line) indicates the bimodality coefficient associated with a uniform distribution. The data underlying this figure can be found in S6 Data.
(TIF)

**S5 Fig. GA-OX1 and SQ4a do not exhibit strong inhibition on nutrient agar.** Spots of GA-OX1 RFP and SQ4a sfGFP plated side-by-side at high and low densities on nutrient agar. (A) A dense culture of GA-OX1 RFP spotted adjacent to single SQ4a sfGFP colonies. (B) A dense culture of SQ4a sfGFP spotted adjacent to single SQ4a sfGFP colonies. (C) A dense culture of SQ4a sfGFP spotted adjacent to single GA-OX1 RFP colonies.
(TIF)

**S6 Fig. Changes in titer of each strain during all inoculation trials.** Bacterial strain titers before and after inoculation trials. The data underlying this figure can be found in S2, S4, S5, and S6 Data files. (A) GA-OX1 GFP and RFP (cf Figs 2B, 2C, and S1). (B) SQ4a GFPmut3 and GA-OX1 RFP (cf Fig 3C and 3D). (C) SQ4a sfGFP and GA-OX1 RFP (cf S4A and S4B Fig). (D) GA-OX1 sfGFP and SQ4a RFP(cf S4C and S4D Fig).
(TIF)

**S1 Table. Community heterogeneity in competition trials between GA-OX1 sfGFP and GA-OX1 RFP.** Bimodality coefficients, population variances, species-level $F_{st}$, and dip statistics calculated from competition trials between GA-OX1 sfGFP and GA-OX1 RFP.
(XLSX)

**S2 Table. Community heterogeneity in competition trials between SQ4a GFPmut3 and GA-OX1 RFP.** Bimodality coefficients, population variances, species-level $F_{st}$, and dip statistics calculated from competition trials between SQ4a GFPmut3 and GA-OX1 RFP.
(XLSX)

**S3 Table. Community heterogeneity in competition trials between SQ4a sfGFP and GA-OX1 RFP.** Bimodality coefficients, population variances, species-level $F_{st}$, and dip statistics calculated from competition trials between GA-OX1 RFP and SQ4a sfGFP.
(XLSX)

**S4 Table. Community heterogeneity in competition trials between SQ4a RFP and GA-OX1 sfGFP.** Bimodality coefficients, population variances, species-level Fst, and dip statistics calculated from competition trials between GA-OX1 GFP and SQ4a RFP.
(XLSX)

**S1 Data. Bacterial symbiont community composition in wild squash bug populations.** Sequence counts of the top 20 *Caballeronia* 16s V3-V4 amplicon sequence variants (ASVs) from 9 bugs across 3 field localities in Georgia, Indiana, and North Carolina. Publicly available data from [7].
(XLSX)

**S2 Data. Insect inoculation trials using *Caballeronia zhejiangensis* GA-OX1 sfGFP and GA-OX1 RFP.** Colony count data of GA-OX1 sfGFP and GA-OX1 RFP colonization trials from each insect, including insects from which no fluorescent colonies were recovered.
(XLSX)

**S3 Data. Competition between *Caballeronia zhejiangensis* GA-OX1 and *Caballeronia* sp. SQ4a.** Colony count data of counterlabeled GA-OX1 and SQ4a strains grown in liquid coculture over 24 h, including both same-strain and interspecific combinations.
(XLSX)

**S4 Data. All bugs dissected in colonization trials using SQ4a GFPmut3 and GA-OX1 RFP.** Colony count data of SQ4a GFPmut3 and GA-OX1 RFP colonization trials from each insect, including insects from which no fluorescent colonies were recovered.
(XLSX)

**S5 Data. All bugs dissected in colonization trials using SQ4a sfGFP and GA-OX1 RFP.** Colony count data of SQ4a sfGFP and GA-OX1 RFP colonization trials from each insect, including insects from which no fluorescent colonies were recovered.
(XLSX)

**S6 Data. All bugs dissected in colonization trials using SQ4a RFP and GA-OX1 sfGFP.** Colony count data of SQ4a RFP and GA-OX1 GFPmut3 colonization trials from each insect, including insects from which no fluorescent colonies were recovered.
(XLSX)

**S7 Data. RFP and GFP fluorescence in bugs co-colonized with GA-OX1 sfGFP and SQ4a RFP.** Red and green fluorescence intensities measured along digital transects of dissected symbiotic organs (M4s).
(ZIP)

**S1 Code. Custom R code for Fig 1C, accompanying S1 Data.**
(R)

**S2 Code. Custom R code for Fig 2B and Fig 2C, accompanying S2 Data.**
(R)

**S3 Code. Custom R code for Fig 3A, accompanying S3 Data.**
(R)

**S4 Code. Custom R code for Fig 3C and 3D, accompanying S4 Data.**
(R)

**S5 Code. Custom R code for Fig 4C, accompanying S7 Data.**
(R)

**S6 Code. Custom R code for S1 Fig, accompanying S2 Data.**
(R)

**S7 Code. Custom R code for S4A Fig and S4B Fig, accompanying S5 Data.**
(R)

**S8 Code. Custom R code for S4C Fig and S4D Fig, accompanying S6 Data.**
(R)

**S9 Code. Custom R code for S6 Fig, accompanying S2 Data, S4 Data, S5 Data, and S6 Data.**
(R)

## Acknowledgments

We thank Gerardo, de Roode, Vega, and Levin lab members for helpful comments on this manuscript. In particular, we wish to thank Joselyne Chavez for supplying ASV diversity data from squash bugs; Anthony Junker for important advice on microscopy and image analysis; Sandra Mendiola and Erik Edwards for maintaining squash bug lab colony stocks and squash plants; and Kayla Stoy, Justine Garcia, and Patrick Stillson for the isolation and genomic characterization of *Caballeronia* strains used in this study. We also thank Travis Wiles and Elena Wall for contributing the plasmids that made this project possible.

## Author Contributions

**Conceptualization:** Jason Z. Chen, Nicole M. Gerardo, Nic M. Vega.

**Data curation:** Jason Z. Chen, Zeeyong Kwong.

**Formal analysis:** Jason Z. Chen, Zeeyong Kwong.

**Funding acquisition:** Nicole M. Gerardo, Nic M. Vega.

**Investigation:** Jason Z. Chen, Zeeyong Kwong.

**Methodology:** Jason Z. Chen, Zeeyong Kwong.

**Project administration:** Jason Z. Chen, Nicole M. Gerardo, Nic M. Vega.

**Resources:** Nicole M. Gerardo, Nic M. Vega.

**Supervision:** Nicole M. Gerardo, Nic M. Vega.

**Visualization:** Jason Z. Chen, Nicole M. Gerardo.

**Writing – original draft:** Jason Z. Chen.

**Writing – review & editing:** Jason Z. Chen, Nicole M. Gerardo, Nic M. Vega.

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
