## [Editor Report · Decision Letter 0]

16 Aug 2023

Dear Dr Chen, 

Thank you for submitting your manuscript entitled "Ecological drift during colonization drives within- and between-host heterogeneity in animal symbiont populations" for consideration as a Research Article by PLOS Biology.

Your manuscript has now been evaluated by the PLOS Biology editorial staff, as well as by an academic editor with relevant expertise, and I'm writing to let you know that we would like to send your submission out for external peer review.

Once your full submission is complete, your paper will undergo a series of checks in preparation for peer review. After your manuscript has passed the checks it will be sent out for review. To provide the metadata for your submission, please Login to Editorial Manager (https://www.editorialmanager.com/pbiology) within two working days, i.e. by Aug 18 2023 11:59PM.

Kind regards,

Roli Roberts

Roland Roberts, PhD

Senior Editor

PLOS Biology

rroberts@plos.org

---

## [Decision Letter · Decision Letter 1]

4 Oct 2023

Dear Dr Chen,

Thank you for your patience while your manuscript "Ecological drift during colonization drives within- and between-host heterogeneity in animal symbiont populations" was peer-reviewed at PLOS Biology. It has now been evaluated by the PLOS Biology editors, an Academic Editor with relevant expertise, and by three independent reviewers. 

In light of the reviews, which you will find at the end of this email, we would like to invite you to revise the work to thoroughly address the reviewers' reports.

You’ll see that reviewer #1 is broadly positive, saying that although the findings are not novel per se, your study represents “a great empirical demonstration of the importance of stochastic processes.” However, s/he raises a number of concerns that may stem from lack of clarity over what you’ve actually done (e.g. re within-host vs between-host differences); s/he also wants a better connection to the literature. Reviewer #2 is also positive, but wants further detail and context (and better framing). S/he thinks that you are over-stating and over-generalising a number of points, that you need to tighten up definitions, run some sort of robustness analysis for the bimodality coefficient, clarify some discrepancies, provide data and code (we would insist on that anyway), and discuss factors that might impact the phenomena described. Reviewer #3 is also positive, but more guardedly so. S/he thinks the paper would benefit from a “conceptual model figure” early on, and makes a number of points, some of which involve options to either do more experiments (do mixed colonisation assays, check later instars) or tone down the claims. The Academic Editor asked me to emphasise the shared concern from the reviewers that you have a tendency to over-generalise your findings.

Given the extent of revision needed, we cannot make a decision about publication until we have seen the revised manuscript and your response to the reviewers' comments. Your revised manuscript is likely to be sent for further evaluation by all or a subset of the reviewers.

**IMPORTANT - SUBMITTING YOUR REVISION**

*Re-submission Checklist*

*Published Peer Review*

*PLOS Data Policy*

*Blot and Gel Data Policy*

Sincerely,

Roli Roberts

Roland Roberts, PhD

Senior Editor

PLOS Biology

rroberts@plos.org

REVIEWERS' COMMENTS:

Reviewer #1:

Dissecting the causes of variation in microbiome composition across hosts is of broad importance. With this study, the authors highlight the role of stochastic processes in leading to variation in strain composition both across and within hosts. With a simple experimental design and beautiful imaging, they show that heterogeneity can be generated across individuals of a host species due to chance differences in colonization of two isogenic strains (other than due to differential markers). Most interestingly they show that chance processes can overcome a clear difference in non-isogenic strains, if the inoculum density is 1000 cfu/ul or lower. Additionally, they show that variation can be maintained within an insect host due to differential colonization of the sub-structures (crypts) of the organ that houses the symbionts within the insect gut.

Overall, I found this an interesting and citable study. My key concern is that the authors do not reconcile their findings of within-host diversity housed in the crypts with the pattern of one strain stochastically dominating in individual hosts. They must have selectively chosen co-colonized hosts; however, I can't find this explained in the methods or results. Additionally, there were a few other places in the manuscript where I was unclear about which marker combinations were being presented and how representative the results were. In Figure 3A, are the in vitro results (n=10) from 1 marker combination or pooled across different ones? Further, in the in vivo study, while the bimodal outcome shown in Fig 3C was reproduceable with different marker constructs, the increase in bimodality with the decrease in inoculum dose (Fig 3D) was not, and some of the results seemed strangely skewed (see Figure S4), so it would be nice if this was unpacked a little (even if only in the supplement). 

One of the strengths of this study is that it is motivated by explaining strain diversity observed among hosts in nature. With that in mind, the manuscript could have benefited from discussing the experimental findings in light of the patterns seen in nature. For example, how do the inoculum doses given in the experiments compare to those bugs would be exposed to in feces? Can this be related back to the coinfections observed in 3C? 

Otherwise, this manuscript was well presented and thoroughly discussed. It is of value to the literature not so much for its novelty, but because it raises to the forefront and provides a great empirical demonstration of the importance of stochastic processes. As such, there are some places where the text (and the citation lists) could be streamlined. Below I flag some places that were confusing or could be improved.

3: The phrase "providing the raw material for natural selection…." is somewhat distracting for setting the stage and could be deleted

8: better to provide a definition of drift here—saying it "causes stochastics fluctuations" is somewhat confusing

9: add in "we found" after the comma

21-29: This paragraph was somewhat awkwardly constructed and could be mostly deleted. 

145-146: Walk the reader through the results a little more. What would the bimodality coefficient be if the data were normally distributed or if it were skewed to one pole?

148: Here do you mean to say that there is "more heterogeneity seen across individual insects"? 

151-152: replace "in separate" with "across"?

166/Figure 3A: Does the n=10 and what is shown depict two different marker schemes or a specific combination?

208: Tables S2-S4 vs Table S1. This result could be explained more.

212-214: clarify that you mean heterogeneity among crypts

218: clarify that you mean heterogeneity among crypts

255: delete "between hosts" or move to previous line

259: remove citations 

288-294: I found this interesting, but it is a fair amount of discussion for something not examined in the manuscript.

337-339: It's not really clear from what you've written how these results vary to the results found in vitro or why host control would lead to coinfection in anterior crypts.

378-387: This information would better go in the legend for Supplemental Figure 3 as the data is not presented in the main manuscript.

391-396: It's not clear whether this information is needed. Was the rifampicin resistant mutant in this study? Shouldn't the GFPmut3 be listed in Table 1? 

434: Are the 4 combinations the ones shown in Fig S6? From the wording here, it sounds like you have both species in self vs. self pairings and then variably labeled in the cross species pairings?

435: Clarify that self vs. self was only done with GA-OX1, not with both strains.

452: what volume of 1x PBS

457: Was the blue dye used in both feeding solutions?

460: Figure S4 does not appear to be the correct figure here

523: Where the Fst presented in the main text? Perhaps you could mention and interpret the Fst values in each section of the results. Or here make a general conclusion about how it relates to the other metrics.

528: It's not clear what the 3 p-values refers to.

898: Should refer to figure 2B? Although the number of points don't seem to match what is shown in 2B?

916-917: These lines should be deleted and replaced by text that helps to explain the bottom 3 rows of the figure. 

920: Provide more information here on how the phylogeny was constructed.

Reviewer #2:

In this study, the authors describe an elegant set of experiments examining the role of ecological drift in the colonization of the M4 organ of the squash bug Anasa tristis by Caballeronia symbionts. Using controlled experiments with fluorescently labeled Caballeronia strains, they find that variation in community composition between hosts can arise as a result of stochastic colonization dynamics and influenced by inoculum size. Interestingly, they also find that the strength of this effect varies spatially along the length of the digestive tract within a host. Overall, I think this study is interesting and may be worthy of publication in PLOS Biology, but some additional detail and context on the authors' results, along with some adjustments to the framing of the manuscript, would greatly strengthen the publication. 

Major comments: 

1. Some of the authors' claims are over-stated and over-generalized. For example: 

 -In several places it is implied that the authors results' are generalizable beyond the squash bug system. For example in lines 51-53: "To illustrate the importance of ecological drift during the establishment of even highly specific symbioses, we employ the squash bug, Anasa tristis (Fig 1A), as a model." In fact, the delayed colonization of the organism and the generalist ability of the Caballeronia to live in both host-associated and free-living environments could both be important factors influencing the size of the effect observed by the authors. I recommend revising the text to limit over-generalizing and in particular editing the title to include the name of the study species.

 -Relatedly, please clarify in the abstract that the focus is on close host-microbe symbioses rather than all host-microbiome interactions. 

 -Lines 12-14 of the abstract: How do the authors' results show that ecological drift is responsible for the within-host heterogeneity, as opposed to other possible host contributors? 

2. At the same time, for the authors' main results on colonization outcomes in individual insects, it is not very clear what an alternative outcome would look like - increasing bimodality seems basically inevitable with shrinking input population sizes. It would be helpful if the authors could provide additional context on what specific alternative hypotheses could be considered, and how the expected results of their experiments might differ under an alternative hypothesis. 

3. Please make sure all terms are clearly and specifically defined in the introduction. For example, the authors seem to use "composition" to refer to microbial taxonomic composition, or which species/strain is dominant (as opposed to other ways we could describe the composition of a microbial community). I would probably also define "heterogeneity" as the authors are using it here.

4. The bimodality coefficient is a nice statistic to quantify the stochasticity the authors are looking for, but it's not clear how robust this metric is to differences in sample size and individual outliers. I recommend the authors perform a bootstrapping or subsampling analysis on their data in order generate confidence intervals/error bars for each of their bimodality coefficient estimates. 

5. Figure 1C needs to have a taxonomy legend. As is, it is not possible to determine what taxa are shown and whether it supports the authors' point.

6. There are some inconsistent or puzzling results reported that are not addressed in the text. Specifically, what do the authors make of the variability in the fraction of colonized insects, and the sometimes sizable share of animals that are not colonized in any single experiment? Secondly, it appears that increasing bimodality with inoculum dose was not observed in Figure S4A-B. What might explain this discrepancy? 

7. I have several follow-up questions about factors that might influence the observed phenomena. I don't believe the authors necessarily need to experimentally test all of these factors prior to publication, but if they have explored or considered the effects of any of these, it would be useful to include and/or mention in the discussion. 

 -Are these bacteria motile? Could there be transmission between crypts? Does the duration of the experiment influence

---

## [Decision Letter · Decision Letter 2]

1 Mar 2024

Dear Dr Chen,

Thank you for your patience while we considered your revised manuscript "Ecological drift during colonization drives within- and between-host heterogeneity in animal-associated symbiont populations" for publication as a Research Article at PLOS Biology. This revised version of your manuscript has been evaluated by the PLOS Biology editors, the Academic Editor and the original reviewers.

Based on the reviews, we are likely to accept this manuscript for publication, provided you satisfactorily address the following data and other policy-related requests.

IMPORTANT - Please attend to the following:

a) Please change your Title to the following: "Ecological drift during colonization drives within-host and between-host heterogeneity in an animal-associated symbiont" (for clarity, and to avoid over-generalisation)

b) Please attend to the remaining very minor points from reviewer #1.

c) Please address my Data Policy requests below; specifically, we need you to supply the numerical values underlying Figs 1C, 2BC, 3ACD, 4C, S1ABC, S3 (treefile), S4ABCD, S6ABCD, either as a supplementary data file or as a permanent DOI’d deposition. I note that you have already submitted a large amount of fairly raw-looking data in your supplementary data files, but its relationship to the individual Figure panels is unclear. Please can you clarify and/or supply files containing the values displayed in the Figs?

d) Please cite the location of the data clearly in all relevant main and supplementary Figure legends, e.g. “The data underlying this Figure can be found in S1 Data” or “The data underlying this Figure can be found in https://doi.org/10.5281/zenodo.XXXXX”

e) Please make any custom code available, either as a supplementary file or as part of your data deposition. Again, I see that you've supplied a number of R scripts as supplementary files - many thanks - so this is merely to check they should suffice to recreate the findings.

We expect to receive your revised manuscript within two weeks. 

*Published Peer Review History*

*Press*

Sincerely,

Roli Roberts

Roland Roberts, PhD

Senior Editor

rroberts@plos.org

PLOS Biology

****[DELETE AS APPROPRIATE]****

ETHICS STATEMENT:

-- Please include the full name of the IACUC/ethics committee that reviewed and approved the animal care and use protocol/permit/project license. Please also include an approval number.

-- Please include the specific national or international regulations/guidelines to which your animal care and use protocol adhered. Please note that institutional or accreditation organization guidelines (such as AAALAC) do not meet this requirement.

-- Please include information about the form of consent (written/oral) given for research involving human participants. All research involving human participants must have been approved by the authors' Institutional Review Board (IRB) or an equivalent committee, and must have been conducted according to the principles expressed in the Declaration of Helsinki.

DATA POLICY:

Regardless of the method selected, please ensure that you provide the individual numerical values that underlie the summary data displayed in the following figure panels as they are essential for readers to assess your analysis and to reproduce it: Figs 1C, 2BC, 3ACD, 4C, S1ABC, S3 (treefile), S4ABCD, S6ABCD. NOTE: the numerical data provided should include all replicates AND the way in which the plotted mean and errors were derived (it should not present only the mean/average values).

DATA NOT SHOWN?

REVIEWERS' COMMENTS:

Reviewer #1:

This is my second time reviewing this manuscript. The authors have done an excellent job addressing comments of all the reviewers. The revised manuscript reads really well now. I also like the changes they have made to the figures and I assume the image quality will be higher upon publication. Additionally, Figure 1 D&E are not cited in the text and their legend incorrectly identifies the colors.

Reviewer #2:

I thank the authors for their thorough response to the reviewers' comments. I believe the manuscript has been substantially clarified and improved and I have no further recommendations.

---

## [Editor Report · Decision Letter 3]

26 Mar 2024

Dear Dr Chen,

Thank you for the submission of your revised Research Article "Ecological drift during colonization drives within-host and between-host heterogeneity in an animal-associated symbiont" for publication in PLOS Biology. On behalf of my colleagues and the Academic Editor, Sebastian Winter, I'm pleased to say that we can in principle accept your manuscript for publication, provided you address any remaining formatting and reporting issues. These will be detailed in an email you should receive within 2-3 business days from our colleagues in the journal operations team; no action is required from you until then. Please note that we will not be able to formally accept your manuscript and schedule it for publication until you have completed any requested changes.

Sincerely,

Roli Roberts

Senior Editor

PLOS Biology

rroberts@plos.org